# Adolescent's descriptions of fatigue, fluctuation and payback in chronic fatigue syndrome/myalgic encephalopathy (CFS/ME): interviews with adolescents and parents

Roxanne M Parslow,[1] Nina Anderson,[1] Danielle Byrne,[1] Alison Shaw,[2] Kirstie L Haywood,[3] Esther Crawley[1]

► Additional material is published online only. To view please visit the journal online (http://dx.doi.org/10.1136/ bmjpo-2018-000281).

[1]Centre for Academic Child Health, University of Bristol, Bristol, UK
[2]Centre for Primary Care Research, Bristol Medical School, University of Bristol, Bristol, UK
[3]Warwick Research in Nursing, Division of Health Sciences, Warwick Medical School, University of Warwick, Health Sciences, Coventry, West Midlands, UK

**Correspondence to**
Professor Esther Crawley;
esther.crawley@bristol.ac.uk

## ABSTRACT

**Objective** As part of a larger qualitative study to explore outcomes important in paediatric chronic fatigue syndrome/myalgic encephalopathy (CFS/ME) and what improvements in fatigue and disability are key, interviews were undertaken with adolescents and their parents. This paper focuses on their descriptions of fatigue, fluctuation of symptoms and payback.

**Design and setting** Semistructured qualitative interviews were undertaken between December 2014 and February 2015. Adolescents and parents were interviewed separately. Participants were recruited from a single specialist paediatric chronic fatigue service. Interviews were audio recorded, transcribed verbatim and analysed using thematic analysis.

**Participants** We interviewed 21 adolescents and their parents (20 mothers and 2 fathers). The adolescents were aged between 12 and 17 years of age (mean age 14.4 years), mild to moderately affected by CFS/ME (not housebound) and the majority were female (16/21).

**Results** All adolescents with CFS/ME reported fatigue, a natural fluctuation of the condition, with good days and bad days as well as an increase in symptoms after activity (payback). However, adolescent's descriptions of fatigue, symptoms and the associated impact on their daily lives differed. The variations included: fatigue versus a collection of symptoms, constant versus variable symptoms and variable symptom severity. There were differences between participants in the amount of activity taken to cause payback. The impact of fatigue and symptoms on function ranged from: limiting the duration and amount of leisure activities, struggling with daily activities (eg, self-care) to no activity (sedentary).

**Conclusions** Fatigue, fluctuation of the condition and payback after activity are described by all adolescents with CFS/ME in this study. However, the individual experience in terms of how they describe it and the degree and impact varies.

## What is already known on this topic?

► Paediatric chronic fatigue syndrome/myalgic encephalopathy (CFS/ME) is relatively common peaking in adolescents between 10 and 19 years of age and disrupts an important time in their development.
► Fatigue and payback are defining features of the condition and these are well described in adults.
► There are no studies describing adolescent's experiences of fatigue and payback in detail.

## What this study hopes to add?

► All adolescents describe unique aspects of fatigue in CFS/ME; how it fluctuates naturally day to day but can also get worse following activity (payback).
► The individual experience of fatigue varies in severity, frequency, the amount of activity taken to cause payback as well as the resulting impact on function.
► The variation in the experience of fatigue needs to be taken into account in treatment.

disabling fatigue and can include diverse physical and cognitive symptoms.[1 2] Paediatric CFS/ME is relatively common with a prevalence of 0.6%–2.4%,[3–8] and the incidence reported to peak in adolescents (10–19 years of age).[9] CFS/ME disrupts the lives of young people; some can become bedbound,[10] miss school[11–13] and stop seeing friends.[14–16]

Fatigue is the defining feature of CFS/ME and 'postexertional malaise' or 'payback' ('physically drained after mild activity')[17] is a universal symptom[18] and key to discriminating patients with CFS/ME from other patient populations.[19] However, CFS/ME is heterogeneous; in a recent study, adult patients with CFS/ME were found to have one of six symptom-based profiles (phenotypes).[18] The concepts of 'fatigue' and

## INTRODUCTION

Chronic fatigue syndrome/myalgic encephalopathy (CFS/ME) is characterised by severe

'payback' are complex as a range of aspects are currently described and measured in CFS/ME and there is little agreement on which are most important.[20 21] Consensus is needed in order to be able to accurately assess the impact of fatigue and payback on patients and measure improvement from treatment. Recently, adults with CFS/ME described the nature of fatigue as three distinct domains: tiredness, fatigue and exhaustion.[22] However, there is less qualitative evidence in children. As part of a larger study investigating outcomes important in paediatric CFS/ME,[23] detailed data arose relating to fatigue, fluctuation of symptoms and payback and this paper aims to describe adolescents' experiences and descriptions of these domains.

## METHODS
### Study design
As part of a larger qualitative study exploring outcomes important in paediatric CFS/ME, semistructured interviews were undertaken with adolescents with CFS/ME and their parents (separately) between December 2014 and February 2015. We felt both perspectives were important to understand the overall impact of CFS/ME.

### Participants
Adolescents aged between 12 and 17 years, diagnosed with mild to moderate CFS/ME (not housebound)[2] and their parents, were recruited from a specialist paediatric chronic fatigue service in South West England. We aimed to recruit a range of participants (age, gender and disease severity) through maximum variation purposive sampling.[24] Families were approached in outpatient clinics, given the appropriate patient information sheets and later contacted by a researcher to arrange an interview. Written consent was obtained from both parents and adolescents in person, prior to the start of the interview.

### Data collection
Previous work exploring how quality of life is impacted by a childhood disorder was reviewed to help inform the development of the topic guide.[25–28] The guide included questions on the overall impact of CFS/ME, how participants would know if they were better/worse or recovered and incorporated a card ranking exercise (online supplementary appendix A). Participants were asked to rank 15 different areas of life impacted by having CFS/ME identified through previous work,[15 29] in order of importance and discuss each card in further detail. Fatigue, fluctuation and payback were previously identified as important to children with CFS/ME[15 29] and therefore included in the card ranking exercise. This paper analyses the detailed qualitative data collected in these areas. Adolescents were mostly interviewed alone; however, in four interviews, a parent was present in the room when their child was interviewed. Interviews were undertaken in participant's own homes and one in the hospital.

### Data analysis
Interviews were recorded on an encrypted digital audio recorder and transcribed verbatim. Transcripts were checked for accuracy, anonymised and uploaded to NVIVO 10[30] to help organise the data and undertake thematic analysis to report patterns (themes) within data.[31] Transcripts for the subgroups (adolescents vs parents and mild-moderate severity) were coded separately in NVIVO and the groups were then compared to search for similarities and differences in the data.[32 33]

Data analysis started using deductive (anticipated) codes of the health outcomes outlined in the card ranking exercise identified from previous research. Transcripts were then read and re-read to further develop inductive (emergent) codes using participants' own language of their descriptions of fatigue, symptoms, fluctuation and payback in detail, producing a long list of codes (online supplementary appendix B). The coding was reviewed as interviews progressed merging or separating codes based on participants' descriptions.[34] Interviews continued until we reached data saturation where new interviews produced little or no change to the inductive codes.[35] Ten transcripts (five adolescents, five parents) were double coded (NA) and compared to improve the trustworthiness of the analyses.[36]

## RESULTS
### Participants
We interviewed 43 participants: 21 adolescents (12–17 years old, mean 14.4 years) and their parents (20 mothers and 2 fathers—2 parents from 1 family were interviewed). The majority of adolescents were female (16/21) and mildly affected by CFS/ME (17/21).[2] Five families initially approached did not participate due to lack of time or being too ill at the time.

### Summary of findings
All adolescents with CFS/ME report fatigue, a natural fluctuation of the condition, as well as an increase in fatigue and symptoms after activity (payback). However, the experience was highly variable between participants. The results describe: (1) adolescents descriptions of fatigue (fatigue vs collection of symptoms, constant vs variable symptoms and variable symptom severity); (2) fluctuation of fatigue and symptoms (good days and bad days); (3) increase in fatigue and symptoms following activity (payback) and (4) the impact of fatigue and symptoms on function (limiting leisure activities, reduced daily activities and becoming sedentary).

### Adolescents descriptions of fatigue
All adolescents reported tiredness and a lack of energy. Adolescents mostly used the term 'tired', but also described 'drowsiness' feeling 'worn out', 'heavy' or 'drained'. Problems sleeping and difficulties waking up were key features associated with fatigue, 'waking up is the hardest bit' (C10, female 13).

I would wake up and I would just feel like I'm dead, like just like I can't get up out of bed. It's horrible. (C17, male 17)

Parents confirmed that mornings were problematic:

…it was a struggle to get her up big, big struggle to get her up. We would be in and out of her bedroom for ages trying to wake her up, or keep her awake. (P19, mother of female 15).

### Fatigue versus a collection of symptoms

Some adolescents described only experiencing 'tiredness', whereas others described a collection of severe symptoms:

There's no really symptoms of it for me, there's no pain or anything, apart from tiredness. (C16, male, 12)

…it's just when it's a combination of being tired, headache, feeling sick, like achy, then it's just like exhausting. (C7, female 17)

Some adolescents felt tiredness was the most bothersome symptom, whereas for others, symptoms such as headaches and feeling sick had the most significant impact:

Yeah, I'd be able to live with my headaches…It's really just exhaustion they stop, it stops you doing things. (C1, female 14)

Probably getting rid of the headaches and the sickness, because the tiredness isn't too bad, but constantly feeling sick is really annoying. (C11, female 16)

### Constant versus variable symptoms

How often symptoms occurred differed between participants as well as changing on a daily basis. Some adolescents described how certain symptoms were constant, 'Achiness, headaches are the main two that they are constant' (C7, female 17). Tiredness was mostly constant, 'tired all the time' (C5, male 12). Parents reflected on this too, 'he was just so tired all the time' (P16, mother of male 12). However, for some adolescents it varied, 'sometimes I'm tired, sometimes I'm not' (C16, male 12).

### Variable symptom severity

Adolescents and their parents often talked about how some symptoms would change in how severe they were. Symptoms were described as being: 'really bad', 'really worn out'. For some, all symptoms became worse with severe tiredness, 'symptoms are like, not as bad as they are when I'm really tired.' (C3, female 13). Adolescents

could articulate different levels of tiredness and the effect this had on their ability to do things:

…like there can be a tiredness where like you just you don't want to get up, and there can be one where you just like you can't do much stuff. So you can be tired and you can be really tired. (C12, male 13).

…she calls it her normal tiredness and her bad tiredness. (P9, mother of female 13)

### Fluctuation of fatigue and symptoms

Adolescents and parents reflected on how CFS/ME naturally fluctuates. CFS/ME was felt to be 'unpredictable'. Parents described the variation in their child's symptoms 'he might have one minor symptom on a good day.' (P12, mother of male 13) and physical function, '1 day she could be going out for a walk then the next day she might asleep' (P6, mother of female 15). Adolescents and parents felt that fluctuation occurred independent of a trigger or overexertion the previous day and found the unpredictability bothersome:

I think it does fluctuate on its own… so yesterday was obviously a bad day, but the day before that I hadn't done anything kind of heavy. (C18, female, 17)

it's not like we pushed too hard. (P1, mother of female 14).

### Good versus bad days

Adolescents and parents recognised patterns of good and bad days:

I have good days most of the week, and then feel terrible for about 3 days. (C12, male 13)

She's not been well, she doesn't really have many good days. (P6, mother of female 15).

Adolescents described bad days where there was an increase in symptoms and tiredness, they found it harder to do things and they got low in mood: 'the bad days, you just, you can't er, really er, get out of bed' (C2, female 12). Parents felt they were able to instantly recognise a worse day based on adolescent's physical signs and mood, 'All I have to do is look at her, I can just tell in her eyes.' (P1, mother of female 14).

In contrast to bad days, adolescents painted a wholly different picture of how they felt on good days, such as having energy, less severe symptoms, an increase in activities: 'I do go to school more when I'm feeling better' (C4, female 13). Adolescents recognised that they felt happier and were more sociable on better days, 'I'll be all happy and I'll be downstairs and talking to everyone non-stop…' (C19, female 15) and parents described their

child with CFS/ME as having two different personalities which allow them to tell what type of day they are having 'when she's feeling better she's a different child' (P9 mother of female 13).

### Increase in fatigue and symptoms following activity (payback)

As well as the natural fluctuation of the condition, adolescents with CFS/ME and their parents talked about the 'payback' that can occur after activity. All adolescents described an extreme level of tiredness and feeling ill with CFS/ME that can occur after activity, 'but if I do any type of activity I will be worse the next day' (C3, female 13). Payback was described as: 'wiped out', 'absolute crash', 'knocked me out', 'zonked out' and 'out cold'.

There were differences in the amount and type of activity taken to result in payback. Some adolescents felt they got payback from all types of activity: 'but if I do any type of activity I will be worse the next day' (C3, female, aged 13). Other adolescents got payback from managing several activities in 1 day:

> a full day of college, and then driving lesson, my concentration for that hour it's fine, but then it completely knocks me for like that evening. So I just have to go careful really. (C8, female 17).

Adolescents questioned whether enjoyable activities were worth it and were often very bothered by payback; it impacted their desire to participate as they were aware of the potential consequences and therefore, it 'held them back' from doing things.

> I will feel completely wiped out and I won't want to do it again. (C5, male 12).

### The impact of fatigue and symptoms on function
#### Limiting leisure activities

Adolescents were limited in the amount of time they could spend on activities, which ranged from minutes, '5 min' to hours, 'an hour at the most'. One parent commented how her child might be able to do a one off short activity, like 'lift an object' but would struggle to sustain it 'because that's an instantaneous one off thing and the others are maintained exertion' (P20, mother of female 13). Adolescents described how even if they attempted activity, this would be for a short duration and they would suddenly feel limited:

> …I'd had two lessons and I just felt like I just couldn't carry on, I just felt really heavy… (C18, female 17)

As a result, participants reduced school attendance, sport, hobbies and leisure activities due to their physical limitations. They wanted to be able to do what normal teenagers their age were doing and not have to think about how long they could do it for.

> …I can't go out for too long otherwise I will be really tired. (C16, male 12)

### Struggling with basic daily activities

Some adolescents described severe tiredness: 'falling asleep and not being able to keep my eyes open.' (C6, female 15). Tiredness directly impacted on adolescents' movement, mobility and physical ability to do daily activities. Those who were moderately affected at the time of the interview more often described problems with self-care such as washing, 'I can't like have a bath by myself' (C3, female 13) and getting dressed, 'I could hardly even get dressed because I could barely stand up' (C12, male 13) as a direct result of fatigue. Adolescents and parents talked about particular problems with walking. This ranged from being limited in walking up stairs and short distances, to not being able to sustain walking for a prolonged period of time:

> …I like limped every time I walked because like my legs would feel really weak. (C12, male 13)

> I still go out for like walks but I don't go as far as when I'm ill. (C13, female 17).

### No activity (sedentary)

When fatigue and symptoms were worse, this resulted in adolescents being very sedentary, 'I would just sit in bed or on the sofa and watch TV for the whole day, because that's all that I could do.' (C16, male 12). Parents reflected on how their child would: 'just sit around' and they often had to physically help their child, 'we were carrying her up the stairs'. Due to their physical limitations, adolescents often did not leave the house:

> have to stay at home probably most of the time on the sofa asleep rather than doing anything else. (C11, female 16)

Adolescents felt like life was 'dull' as they were always tired and could not enjoy activities: 'nothing is exciting or fun anymore.' They also described a lack of motivation to do things and withdrawal: 'I will just stay upstairs for the day. Don't really want to socialise.' (C19, female 15).

### DISCUSSION

Fatigue, fluctuation and payback were described by all adolescents with CFS/ME in this study. However, the individual experience in terms of how they describe it and the degree and impact varies. This study highlights key aspects of fatigue described by adolescents with CFS/ME including: sleep problems, problems waking, fluctuation, payback (feeling tired after activity), problems with the duration of activities, daily activities, mobility and being sedentary.

## Strengths and limitations of the study

This is the largest qualitative study of adolescents with CFS/ME and their parents to date (n=43). The sample included adolescents across what may be considered a wide developmental age range. However, fewer males and moderately affected patients were recruited reducing opportunities to explore differences they might experience.[24] Only two fathers were interviewed as mothers were mostly present when recruiting families from outpatient clinics. Further research is needed to explore their experiences. The sample was limited to English speaking families from a single site. Nevertheless, it is the largest specialist service in the UK covering a wide region. Most adolescents were interviewed alone without a parent present which may have reduced socially desirable answers.[37 38] Differences between subgroups were considered in the analysis; no major differences emerged between children and parents when describing fatigue, symptoms and their impact; however, moderately affected children described more problems with self-care. The data analysis was systematic and rigorous; double coding contributed to the checking and interpretation of data by independent researchers.[34 39]

## Previous research

Payback was described by all adolescents and has been found to be a universal symptom in adults with CFS/ME.[18] Similar to adults with CFS/ME who described fatigue as: tiredness (feeling heavy, sleep disturbance), fatigue (eg, waking up tired, reduced daily activities) and exhaustion (eg, physical collapse, 'so tired I fall over'),[22] adolescents in this study described the same domains but 'payback' was used instead of 'exhaustion' and they did not always say 'fatigue' caused sedentary days. Our results are consistent with previous research where children described the intensity of symptoms fluctuating as well as 'overextension' making it worse, resulting in 'paying the price.'[15 40]

This study details the extremely individual and variable nature of CFS/ME. Whereas one teenager may experience constant fatigue and get payback after managing several activities, another may have a range of symptoms that vary and any activity would result in an increase in symptom severity. All experienced good days and bad days appearing better on good days and able to do more, often returning to school or activities. Differences between mild and moderately affected patients were highlighted as those moderately affected described more problems with self-care.

## CONCLUSION

In this study, adolescents described key aspects of fatigue, fluctuation of symptoms and the impact this had on their activity. Paediatricians need to be aware of how adolescents describe the physical experience of CFS/ME.

**Acknowledgements** We would like to thank all the participants who took part in the study. We are grateful for the support of the clinicians at the specialist paediatric CFS/ME service who helped to identify and recruit participants.

**Contributors** RMP, EC, AS and KLH designed the study. RMP, NA and DB collected the data. RMP and NA analysed the data. All authors contributed to the interpretation of results and to drafting this paper. All authors have read and approved the final version of the manuscript.

**Funding** This work was supported by a University of Bristol PhD Scholarship. EC is funded by the National Institute for Health Research (Senior Research Fellowship, SRF-2013- 06-013).

**Disclaimer** The views expressed in this publication are those of the authors and not necessarily those of the NHS, the National Institute for Health Research or the Department of Health.

**Competing interests** EC was the medical advisor for the Association for Young people with ME (AYME) until 2017.

**Patient consent** Not required.

**Ethics approval** Full ethical approval was obtained from the NRES Committee North West (08/04/2014, ref 14/NW/0170). An amendment to use one-to-one interviews with adolescents, parents and clinicians who are unable to attend focus groups and to include exploration of important outcomes was approved (21/10/2014, ref 14/NW/0770). Relevant R&D approval was obtained from the RNHRD (20/06/2014, ref-RBB 427).

**Provenance and peer review** Not commissioned; externally peer reviewed.

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
