## [Reviewer comments · BMJ Paediatrics Open]

ARTICLE DETAILS

TITLE (PROVISIONAL)	Adolescent's descriptions of fatigue, fluctuation and payback in Chronic Fatigue Syndrome/Myalgic Encephalopathy (CFS/ME): Interviews with adolescents and parents
AUTHORS	Parslow, Roxanne; Anderson, Nina; Byrne, Danielle; Shaw, Alison; Haywood, Kirstie; Crawley, Esther

VERSION 1 – REVIEW

REVIEWER	Reviewer name: David vickers Institution and Country: CCS NHS Trust, UK Competing interests: none
REVIEW RETURNED	14-Mar-2018

GENERAL COMMENTS	a great study with some useful insights to young peoples views
--

REVIEWER	Reviewer name: Patricia Lucas Institution and Country: University of Bristol, UK Competing interests: None
REVIEW RETURNED	12-Apr-2018

GENERAL COMMENTS	An interesting qualitative study reporting on the experiences of children and young people with CFS/ME. The study appears to be well conducted and analysed, and authors interpretations and consistently supported by quotes from young people or their parents. My requested revisions mostly pertain to the write up of the study, but also to a need to better situate this study within the literature on CFS/ME and or chronic illness. 1) There appears to me to be a mismatch between the studies stated aim and the findings reported. The aim is "to explore the impact of CFS/ME" but the reporting is largely describing how symptoms are experienced. This is not my field, but extensive reporting of fatigue symptoms did not add much to my (lay) knowledge. 2) The report would benefit from more detail on study design. a. What was the purpose of interviewing both young people and their parents? b. Having decided to interview both parents and children from the same families, how was this considered in analysis? It doesn't appear to have been considered in this report. c. I note that there are 22 parents and 21 young people, which implies two parents from one family were interviewed. It would be useful to be explicit about family groupings, and to explain how the parent to be interviewed was identified. I was left wondering about family structure, and why 2 men (and only 2 men) were interviewed. d. You only mentioned double coding of data in the discussion – move to methods please. It is good practice but lost at the moment. e. Apart from the sorting task, what was included in the topic guides and how were these topics arrived at? How was existing research and theory used to inform these?
---

	f. The analysis plan suggests that only inductive themes were included. Is this correct? It reads to me as if 'payback' was present in both interviews and analysis because of existing research and practice in the field (also see point 4 below). g. I don't understand table 1. Are these improvements following treatment or adaptations to chronic illness? More literature on this topic was needed. h. The discussion says participants were mildly or moderately affected, but I don't know how this was decided nor how these two categories were used in analysis, they are not mentioned in results. 3) I think the discussion overclaims. There is no evidence produced to support that claims in intro and disc about prior small or limited literature. In fact the results are not compared to literature about the experience of adolescence with CFFS/ME at all: there is no reason to exclude quantitative literature, and qualitative literature DOES exist, it just isn't cited here. I don't think an age range of 5 years is 'wide'. The jump to child outcome measures in the conclusions is unsupported by the rest of the paper, and while this MIGHT be considered an implication it is not a conclusion of this research which wasn't about assessing outcomes. 4) I found the place of 'payback' in the whole confusing. It is presented as both something already established, and something emerging from interviews. This can be seen most obviously in 'what is known' and 'what is added' where the explanation and definition appears in the latter not the former. It appears to be something established in the literature, but its place in the words of participants and in the analytic frame is unclear. I was left with the feeling that, in interviews, participants had reproduced what they had been told about their illness, but this was not commented on. More literature on this topic was needed. 5) The statement that CFS is "relatively common" should be substituted with prevalence/incidence estimates and it would be useful if these considered an international audience.
--	---

VERSION 1 – AUTHOR RESPONSE

Reviewer: 1

1) A great study with some useful insights to young people's views.

Thank you.

Reviewer: 2

An interesting qualitative study reporting on the experiences of children and young people with CFS/ME. The study appears to be well conducted and analysed, and authors interpretations and consistently supported by quotes from young people or their parents. My requested revisions mostly pertain to the write up of the study, but also to a need to better situate this study within the literature on CFS/ME and or chronic illness.

Thank you.

1) There appears to me to be a mismatch between the studies stated aim and the findings reported. The aim is "to explore the impact of CFS/ME" but the reporting is largely describing how symptoms are experienced. This is not my field, but extensive reporting of fatigue symptoms did not add much to my (lay) knowledge.

Thank you, we agree. This paper is part of a larger study exploring the impact of paediatric CFS/ME on adolescents. Due to the detailed data we obtained on fatigue and symptoms we wanted to write a paper solely on these findings as they are important topic in current paediatric CFS/ME clinical practice and research. In the abstract we state (page 3):

“As part of a larger qualitative study to explore outcomes important in paediatric CFS/ME and what improvement in fatigue and disability are key, interviews were undertaken with adolescents and their parents. This paper focuses on their descriptions of fatigue, fluctuation of symptoms and payback.”

We have made this clearer in the introduction and stated aim (page 5):

Recently, adults with CFS/ME described the nature of fatigue as three distinct domains: tiredness, fatigue and exhaustion 19. However, there is less qualitative evidence in children. As part of a larger study investigating outcomes important in paediatric CFS/ME, we noticed a large amount of data arising relating to fatigue and symptoms and this paper describes adolescent’s experiences of fatigue, fluctuation of symptoms and payback specifically.

2) The report would benefit from more detail on study design.

a. What was the purpose of interviewing both young people and their parents?

We sought to interview both children and parents for the larger study as both their perspectives were considered important to understand the overall impact of CFS/ME on children and what outcomes are important in recovery.

We have clarified this in the methods (page 6):

“As part of a larger qualitative study exploring outcomes important in paediatric CFS/ME, semi structured interviews were undertaken with adolescents with CFS/ME and their parents (separately). We felt both perspectives were important to understand the overall impact of CFS/ME.”

b. Having decided to interview both parents and children from the same families, how was this considered in analysis? It doesn’t appear to have been considered in this report.

Thank you for highlighting this. We did consider differences between parents and children in our analysis. We coded transcripts for children and parents separately in NVIVO and compared the coding for any similarities or differences. However, no major differences emerged between parents and children when specifically describing fatigue, symptoms and payback for the purposes of this paper. Therefore, quotes from both children and parents are used to illustrate themes in the results.

We have clarified this in the methods- data collection and analysis (page 7):

“Transcripts for the subgroups (adolescents vs parents and mild-moderate severity) were coded separately in NVIVO 23 and the groups were then compared to search for similarities and differences in the data 25,26.”

And in the Discussion (page 19):

“Most adolescents were interviewed alone without a parent present which may have reduced socially desirable answers 29,30. Differences between sub groups were considered in the analysis; no major differences emerged between children and parents when describing fatigue, symptoms and their impact, however, moderately affected children described more problems with self-care. “

c. I note that there are 22 parents and 21 young people, which implies two parents from one family were interviewed. It would be useful to be explicit about family groupings, and to explain how the parent to be interviewed was identified. I was left wondering about family structure, and why 2 men (and only 2 men) were interviewed.

That is correct- for one family both parents were interviewed. We approached families in clinic and young people were more often accompanied by their mothers. In families with CFS/ME, mothers have usually stopped work to provide care for their child and therefore they were available for interview more often than fathers.

This has been clarified in the results- participants (page 9):

“We interviewed 43 participants: 21 adolescents (12-17 years old, mean 14.4 years), and their parents (20 mothers and 2 fathers- two parents from one family were interviewed).”

We have reflected upon the limitation of less fathers in the discussion- strengths and limitations (page 19):

“This is the largest qualitative study of adolescents with CFS/ME and their parents to date (n=43). The sample included adolescents across what may be considered a wide developmental age range. However, fewer males and moderately affected patients were recruited reducing opportunities to explore differences they might experience. Only two fathers were interviewed as mothers were mostly present when recruiting families from outpatient clinics. Further research is needed to explore the experiences of fathers.”

d. You only mentioned double coding of data in the discussion – move to methods please. It is good practice but lost at the moment.

Thank you, we do mention double coding in the methods already (page 7-8):

“Ten transcripts (5 adolescents, 5 parents) were double coded (NA) and compared to improve the trustworthiness of the analyses 28.”

e. Apart from the sorting task, what was included in the topic guides and how were these topics arrived at? How was existing research and theory used to inform these?

We have now included the semi structured interview guide as appendix A and added further detail on how the topic guide was developed. Methods, study design (page 6):

“Previous work exploring how quality of life is impacted by a childhood disorder was reviewed to help inform the development of the topic guide 20-23. The guide included questions on the overall impact of CFS/ME, how participants would know if they were better/ worse or recovered and incorporated a card ranking exercise (Appendix A). Participants were asked to rank 15 different areas of life impacted by having CFS/ME identified through previous work 13,24, in order of importance, and discuss each card in further detail. The draft topic guide was reviewed by a secondary school, Young Person’s Advisory Group (YPAG) to ensure it was age appropriate. For this paper, we analysed data collected when participants specifically talked about fatigue, fluctuation and payback.”

f. The analysis plan suggests that only inductive themes were included. Is this correct? It reads to me as if ‘payback’ was present in both interviews and analysis because of existing research and practice in the field (also see point 4 below).

Thank you for pointing this out. The analysis was both deductive and inductive as the health outcomes used in the card ranking were identified from previous qualitative studies and therefore analysis began more deductively. How participants then described fatigue, fluctuation and payback in more detail was then analysed more inductively.

We have clarified this in the methods- data collection and analysis (page 7):

“Interviews were recorded on an encrypted digital audio recorder and transcribed verbatim. Transcripts were checked for accuracy, anonymised and uploaded to NVIVO 10 27 to help organise the data and undertake thematic analysis to report patterns (themes) within data 28. Data analysis started using deductive (anticipated) codes of the health outcomes outlined in the card ranking exercise identified from previous research. Transcripts were then read and re-read to further develop inductive (emergent) codes using participants’ own language of their descriptions of fatigue, fluctuation and payback in detail producing a long list of codes.”

g. I don’t understand table 1. Are these improvements following treatment or adaptations to chronic illness? More literature on this topic was needed.

This section describes the change in participants description of fatigue, symptoms and payback over time. This is important in understanding the illness. We state (page 16):

“Improvements were often linked to treatment and naturally learning to manage the condition.”

We have tried to make this easier to understand by changing the first sentence so this now reads:

“In addition to the “natural fluctuation” and “increase in symptoms after payback”, participants often described changes over time. Improvements were often linked to treatment and naturally learning to manage the condition.”

h. The discussion says participants were mildly or moderately affected, but I don’t know how this was decided nor how these two categories were used in analysis, they are not mentioned in results.

Mild and moderately affected participants were recruited based on NICE guidance. In the methods we state and reference NICE guidance (methods, page 6):

“Adolescents aged between 12- 17 years, diagnosed with mild to moderate CFS/ME (not housebound) 2 and their parents, were recruited from a specialist paediatric chronic fatigue service in South West England.”

We also mention differences between mildly and moderately affected participants in the results (page 15):

“Those who were moderately affected at the time of the interview more often described problems with self-care such as washing, “I can’t like have a bath by myself” (C3, female 13) and getting dressed, “I could hardly even get dressed because I could barely stand up” (C12, male 13) as a direct result of fatigue.”

We have now further clarified this in the discussion (page 19):

“Differences between sub groups were considered in the analysis; no major differences emerged between children and parents when describing fatigue, symptoms and their impact, however, moderately affected children described more problems with self-care.”

3) I think the discussion overclaims. There is no evidence produced to support that claims in intro and disc about prior small or limited literature. In fact the results are not compared to literature about the experience of adolescence with CFS/ME at all: there is no reason to exclude quantitative literature, and qualitative literature DOES exist, it just isn’t cited here.

We cite papers in the introduction and discussion where fatigue and payback have been described in detail by adults. There have been qualitative papers in children which describe the overall impact of CFS/ME more broadly, which we cite (see below). However, this paper is novel in that it is the first to describe adolescent's experiences of fatigue and payback qualitatively in detail. We have clarified this in the introduction (page 5):

"Recently, adults with CFS/ME described the nature of fatigue as three distinct domains: tiredness, fatigue and exhaustion 22. However, there is less qualitative evidence in children. As part of a larger study investigating outcomes important in paediatric CFS/ME, a large amount of data arose relating to fatigue, symptoms and payback. This paper aims to describe adolescent's experiences of fatigue, fluctuation of symptoms and payback specifically."

Winger A, Kvarstein G, Wyller VB, et al. Health related quality of life in adolescents with chronic fatigue syndrome: a cross-sectional study. *Health and quality of life outcomes*. 2015;13:96.

Taylor RR, O'Brien J, Kielhofner G, Lee SW, Katz B, Mears C. The occupational and quality of life consequences of chronic fatigue syndrome/myalgic encephalomyelitis in young people. *Br J Occup Ther*. 2010;73(11):524-530.

Parslow R, Patel A, Beasant L, Haywood K, Johnson D, Crawley E. What matters to children with CFS/ME? A conceptual model as the first stage in developing a PROM. *Arch Dis Child*. 2015;100(12):1141-1147.

Parslow RM, Harris S, Broughton J, et al. Children's experiences of chronic fatigue syndrome/myalgic encephalomyelitis (CFS/ME): a systematic review and meta-ethnography of qualitative studies. *BMJ open*. 2017;7(1).

I don't think an age range of 5 years is 'wide'.

Whilst this age range may not be wide in adults, the difference between a 12 year old and 17 year old is great in terms of the way they view and describe their illness. We have clarified this by changing this in the discussion- strengths and weaknesses (page 19):

"The sample included adolescents across what may be considered a wide developmental age range."

The jump to child outcome measures in the conclusions is unsupported by the rest of the paper, and while this MIGHT be considered an implication it is not a conclusion of this research which wasn't about assessing outcomes.

The larger qualitative was about assessing outcomes important to adolescents to develop a new outcomes measure. We have made this clearer throughout the paper referring to "outcomes" rather than "impact of paediatric CFS/ME" for example, abstract, page 3:

"As part of a larger qualitative study to explore outcomes important in paediatric CFS/ME and what improvement in fatigue and disability are key, interviews were undertaken with adolescents and their parents. This paper focuses on their descriptions of fatigue, fluctuation of symptoms and payback."

We cite literature (below) in the introduction where a range of aspects of fatigue are described by adults making measurement of outcomes from care difficult. The detailed descriptions we obtained on fatigue and symptoms from adolescents are important for clinical care and should be considered in the development of a new outcome measure.

Jason L, Jessen T, Porter N, Boulton A, Gloria-Njoku M. Examining types of fatigue among individuals with ME/CFS. *Disability Studies Quarterly*. 2009;29(3).

Jason LA, Evans M, Brown M, et al. Fatigue scales and chronic fatigue syndrome: Issues of sensitivity and specificity. *Disability studies quarterly: DSQ*. 2011;31(1).

Dittner AJ, Wessely SC, Brown RG. The assessment of fatigue. A practical guide for clinicians and researchers. *J Psychosom Res*. 2004;56(2):157-170.

4) I found the place of 'payback' in the whole confusing. It is presented as both something already established, and something emerging from interviews. This can be seen most obviously in 'what is known' and 'what is added' where the explanation and definition appears in the latter not the former. It appears to be something established in the literature, but its place in the words of participants and in the analytic frame is unclear. I was left with the feeling that, in interviews, participants had reproduced what they had been told about their illness, but this was not commented on. More literature on this topic was needed.

This is a complicated area. Payback is defined as a core symptom in NHS guidance, and is used in the clinical setting. However, there are few descriptions and minimal qualitative data from young people about what this means for them.

In the discussion, we described the adult literature as the only comparison (page 20):

"Payback was described by all adolescents and has been found to be a universal symptom in adults with CFS/ME 18. In a recent study, adults with CFS/ME described fatigue as three distinct domains: tiredness (feeling heavy, sleep disturbance), fatigue (e.g. waking up tired, reduced daily activities) and exhaustion (e.g. physical collapse, 'so tired I fall over') 22."

This demonstrates there is knowledge about it, but the novelty of our paper is the work in adolescents. We state page 20):

"The same aspects were identified in this study. However the term 'payback' was used instead of exhaustion. Adolescents described instances where they were very sedentary, unable to continue with activities indicating 'fatigue'. Differences between mild and moderately affected patients were highlighted as those moderately affected described more problems with self-care."

We have clarified this in the "what is known about the subject" by making it clearer that we are talking about adults. This now reads (page 2):

"Fatigue and payback are defining features of the condition and these are well described in adults."

5) The statement that CFS is "relatively common" should be substituted with prevalence/incidence estimates and it would be useful if these considered an international audience.

Prevalence figures have now been added to the introduction (page 5):

"Paediatric CFS/ME is relatively common with a prevalence of 0.6-2.4% 3-8, with the incidence reported to peak in adolescents (10-19 years of age) 9."